

# Variation of soil bacterial communities along a chronosequence of *Eucalyptus* plantation

Jiayu Li[1,2], Jiayi Lin[1], Chenyu Pei[1], Kaitao Lai[3], Thomas C. Jeffries[2] and Guangda Tang[1]

[1] College of Forestry and Landscape Architecture, South China Limestone Plants Research Center, South China Agricultural University, Guangzhou, China
[2] School of Science and Health, University of Western Sydney, Penrith, NSW, Australia
[3] Health and Biosecurity, Commonwealth Scientific and Industrial Research Organisation, North Ryde, NSW, Australia

Corresponding authors
Thomas C. Jeffries,
T.Jeffries@westernsydney.edu.au
Guangda Tang, gdtang@scau.edu.cn

## ABSTRACT

*Eucalyptus* is harvested for wood and fiber production in many tropical and sub-tropical habitats globally. Plantation has been controversial because of its influence on the surrounding environment, however, the influence of massive *Eucalyptus* planting on soil microbial communities is unclear. Here we applied high-throughput sequencing of the 16S rRNA gene to assess the microbial community composition and diversity of planting chronosequences, involving two, five and ten years of *Eucalyptus* plantation, comparing to that of secondary-forest in South China. We found that significant changes in the composition of soil bacteria occurred when the forests were converted from secondary-forest to *Eucalyptus*. The bacterial community structure was clearly distinct from control and five year samples after *Eucalyptus* was grown for 2 and 10 years, highlighting the influence of this plantation on local soil microbial communities. These groupings indicated a cycle of impact (2 and 10 year plantations) and low impact (5-year plantations) in this chronosequence of *Eucalyptus* plantation. Community patterns were underpinned by shifts in soil properties such as pH and phosphorus concentration. Concurrently, key soil taxonomic groups such as *Actinobacteria* showed abundance shifts, increasing in impacted plantations and decreasing in low impacted samples. Shifts in taxonomy were reflected in a shift in metabolic potential, including pathways for nutrient cycles such as carbon fixation, which changed in abundance over time following *Eucalyptus* plantation. Combined these results confirm that *Eucalyptus* plantation can change the community structure and diversity of soil microorganisms with strong implications for land-management and maintaining the health of these ecosystems.

## INTRODUCTION

*Eucalyptus* (*Eucalyptus* spp.), a Myrtaceae species that is native to Australia, is now extensively planted at the global scale because of its fast-growth and strong adaptability to the local environment. It occupies approximately 20 million hectares within the tropical artificial forests (*Cook, Binkley & Stape, 2016*), while in China there are nearly 3.7 million

ha of plantation, making the country the second largest *Eucalyptus* plantation area (*Versini et al., 2014*; *Zhang et al., 2016*). Accompanying notable economic benefits (*Ahmed, 1989*), long-term *Eucalyptus* plantation, however, could induce some severe ecological impacts involving soil degradation and understory diversity loss due to allelopathy (*Zhang & Fu, 2009*; *Yang et al., 2017*; *Fang et al., 2009*). Soil degradation via allelopathy likely impacts soil microbial diversity at the community level due to inhibition of key microbial taxa (*Bertin, Yang & Weston, 2003*).

In terrestrial ecosystems, soil microorganisms play crucial roles in ecological processes including pedogenesis, organic matter decomposition and nutrient cycling (*Hu, Chen & He, 2015*; *Zeng, An & Liu, 2017*; *Hu, Xu & He, 2014*). Soil bacteria are the most abundant and functionally diverse microbial taxa that drive processes which mediate soil quality and decompose organic substances (*He et al., 2009*), and the composition and diversity of soil bacterial communities are sensitive to environmental factors like soil characteristics and vegetation type (*Huang, Xu & Chen, 2008*; *Leecruz et al., 2013*). Plant productivity, diversity and community composition are driven directly or indirectly by the strong interdependence between plants and soil microbes, whereby a series of microbial metabolic activities that liberate available nutrients or other compounds can then be acquired by plants (*Heijden, Bardgett & Straalen, 2008*). Conversely, soil bacterial and fungal communities have been shown to be affected by *Eucalyptus* plantations using PLFA (phospholipid fatty acid) analysis since the community structure was impacted significantly with the variation of planting age (*Chen et al., 2013*; *Cao et al., 2010*; *Wu et al., 2013*). This is contradictory to other studies reported which demonstrate that *Eucalyptus* plantations reduced microbial biomass, soil organic carbon and nitrogen concentrations during continuous planting (*Behera & Sahani, 2003*; *Cortez et al., 2014*). Another relevant study of secondary tropical forest (SF) converted into *Eucalyptus* plantations (EP) using high-throughput sequencing techniques revealed a significant difference between SF and EP samples in bacterial composition and diversity (*Lan et al., 2017*). Given that these tools can elucidate the impact of artificial planting, especially on the surrounding soil ecosystem affected by the functioning of microbial communities if impacted (*Zhang et al., 2017*; *Zheng et al., 2017*; *Zhou et al., 2017a*; *Zhou et al., 2017b*; *Lin et al., 2017*), their application will better define the response of soil microbial community diversity and function to land use shift and long-term *Eucalyptus* plantation, which remains obscure.

In this study, we used high-throughput sequencing of the 16S rRNA gene to assess soil bacterial diversity and community composition and shifts in functional potential along a chronosequence of *Eucalyptus* plantations (2 years, 5 years and 10 years; a nearby secondary-forest was simultaneously investigated as control). We highlighted the following questions: (a) how soil bacterial diversity and community composition varied between different *Eucalyptus* plantation stages and the secondary-forest, (b) what are the main factors driving the structure and composition of bacterial communities along the plantation chronosequences and (c) how functional profiles of bacterial communities involving nutrient cycling, metabolism and degradation shifted by *Eucalyptus* plantation at different growing stages.
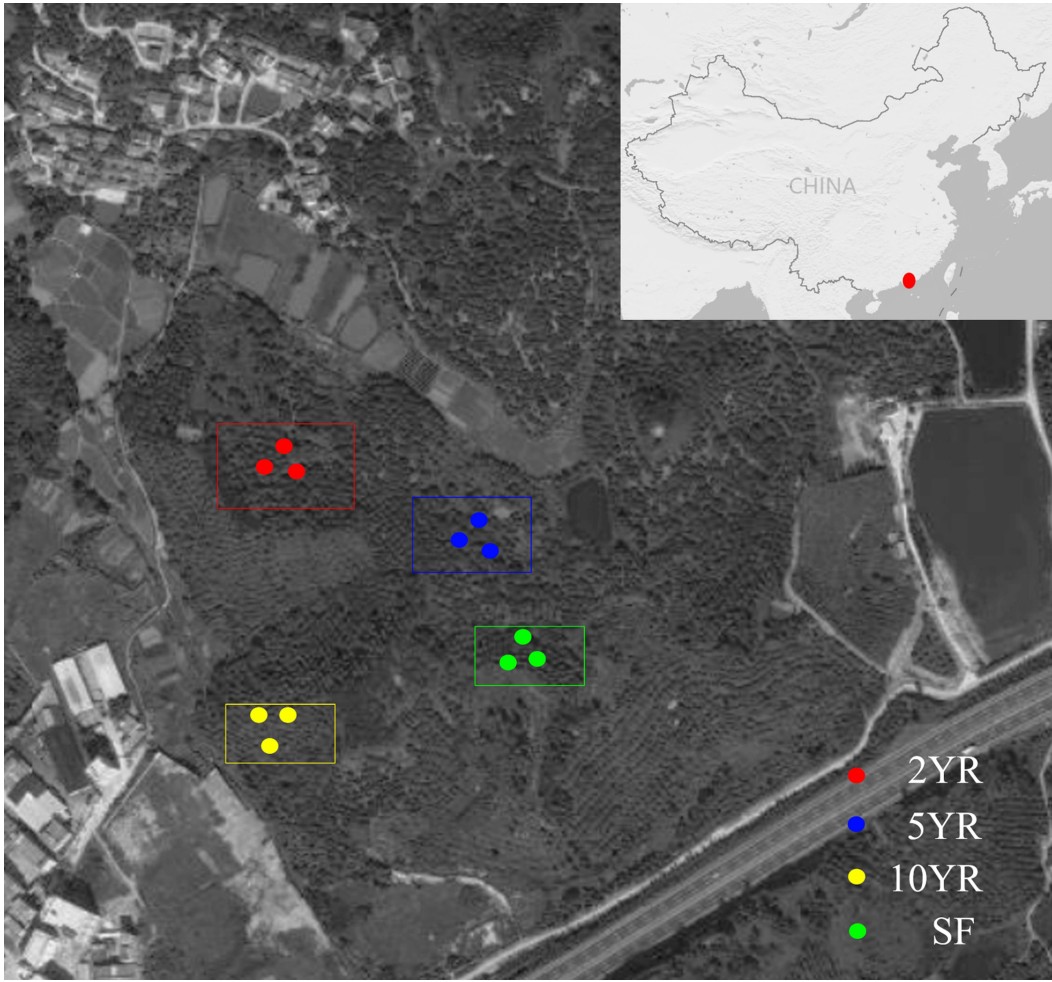

**Figure 1  Sampling design and position in the Zhenlong Town of Huizhou City, Guangdong Province, South China (22°96′N, 114°36′E).**

## MATERIALS & METHODS

### Site information and soil sampling

Soil samples were collected from *Eucalyptus* plantations in the Zhenlong Town of Huizhou City, Guangdong Province, South China (22°96′N, 114°36′E) (Fig. 1) on June 8th, 2017. The climate of this region belongs to the subtropical monsoon climate with an average annual precipitation of 2,000 mm, mainly from April to September, and the mean annual temperature was 22 °C. The current *Eucalyptus* forests have been planted for many years as a substitution after cutting down all trees in previous secondary-forest. An adjacent secondary-forest which was defined as forest lands formed naturally under the impact of human activities (*Corlett, 1994*) was investigated and sampled as the control.

According to the space for time substitution procedure (*Zheng et al., 2017*), we selected four forests including 2 years (2YR), 5 years (5YR) and 10 years (10YR) *Eucalyptus* plantations and the secondary-forest. In each forest, three $20 \times 20$ m² plots were established

in random locations and soil samples were collected. Within each plot, the detritus and litter were swept away, and five soil cores of 0∼15 cm depth was collected, pooled and mixed into a composite sample. A total of 12 composite soil samples were collected from every different *Eucalyptus* plantation stages and secondary-forest on a similar gradient and altitude approximately. Composite soil samples were stored with ice bags and transported to the laboratory as soon as possible for downstream treatments. All the samples were sieved through two mm mesh and divided into two groups, with one group stored in four °C for soil properties measurement, and the other group frozen in −80 °C for soil community DNA extraction.

## Measurement on soil physicochemical properties

The soil pH was determined by a pH monitor with a soil to water ratio of 1:2.5. The soil organic carbon (SOC) and total nitrogen (TN) were measured in $K_2Cr_2O_7$ oxidation and the Kjeldahl method. Total phosphorus (TP) and total potassium (TK) were measured with the alum molybdate yellow colorimetric method and the atomic absorption spectrophotometry. Available nitrogen (AN) and available potassium (AK) were quantified using the alkali-hydrolyzed diffusing method and flame photometer, respectively. Available phosphorus (AP) was immersed with 0.5mol/L $NaHCO_3$, and then measured by Mo-Sb colorimetric method. Soil ammonium nitrogen ($NH_4^+$-N) and nitrate nitrogen ($NO_3^-$-N) were assayed with a flow analyzer (AA3; SEAL analytical, Norderstedt, Germany).

## Soil community DNA extraction and high-throughput sequencing

The total genomic DNA of soil was extracted from 0.2 g of each sample using the Power Soil DNA Isolation kit (MoBio Industries, Carlsbad, CA, USA) following the manufacturer's instructions. Quality of the extracted DNA was determined with a NanoVue Plus spectrophotometer (GE, New Jersey, USA) and 1% agarose gel electrophoresis. The V3-V4 hyper-variable region of the 16S rRNA gene was amplified using primers 338F (5′-ACTCCTACGGGAGGCAGCAG-3′) and 806R (5′-GGACTACHVGGGTWTCTAAT-3′). The following PCR reaction program consists of an initial denaturation at 95 °C for 3 min, 28 cycles of 95 °C for 30 s, 55 °C for 30 s, 72 °C for 45s, then 72 °C for 10 min and 10 °C until halted manually. PCR products were pooled with an equimolar concentration and sequenced with length of 300 bp using a MiSeq sequencer and TruSeq chemistry (Illumina, San Diego, CA, USA) at Shanghai Majorbio Bio-pharm Technology Co., Ltd.

## Bioinformatics analysis

After sequencing, the primary analysis of raw FASTQ data was processed with the QIIME2 pipeline (version 2017.10; http://qiime2.org/) (*Caporaso et al., 2010*). Briefly, DADA2 (*Callahan et al., 2016*) was used for error-correction, quality filtering, chimera removal and sequence variant calling of the Illumina amplicon sequences, with reads truncated at 270 bp, corresponding to a quality score >20. Resultant feature sequences (sOTUs) were summarized and then annotated using an RDP classifier (*Cole et al., 2009*) pre-trained to the full-length Greengenes database (version Aug, 2013) (*DeSantis et al., 2006*). sOTU can be used interchangeably with amplicon sequence variants that it refers to sequences that differ by >1 nucleotide. The predictive functional categories of bacterial community

were annotated through Phylogenetic Investigation of Communities by Reconstruction of Unobserved States (PICRUSt) and generated metabolic pathways depending on KEGG Orthologs at level 3 after importing the normalized sOTU table (*Langille et al., 2013*). This sOTU table was generated in QIIME2 by performing closed reference clustering of sOTU sequence variants against the Greengenes database (version Aug, 2013, 99%) (*DeSantis et al., 2006*) using VSEARCH (*Rognes et al., 2016*). The raw data have been deposited with the European Nucleotide Archive database (http://www.ebi.ac.uk/ena) under accession code ERP109013.

## Statistical analysis

The estimated alpha diversity including number of sOTUs, Shannon and Chao1 indices were calculated within QIIME2 with a resampling depth of 27,838 sequences *per* sample to ensure even sampling depth with statistical significance determined using Kruskal-Wallis tests. Using rarefied sOTU table, the relative correlation between biotic and environmental similarity matrices was calculated using Mantel-tests within QIIME2 using the vegan package of R (*Oksanen et al., 2007*) to elucidate which variables most influenced microbial community structure. Principal coordinates analysis (PCoA) was carried out based on the weighted Unifrac distance (*Lozupone et al., 2011*) between samples in QIIME2 with the significance of sample groupings determined using Analysis of Similarities (ANOSIM) (*Clarke, 1993*). Environmental variables found to have the most significant influence using the Mantel-tests were mapped as colour gradients to PCoA ordinations. According to the results of mantel test, we choose pH ($P = 0.025$, $r = 0.328$), TN ($P = 0.048$, $r = 0.297$), TP ($P = 0.041$, $r = 0.303$) and AP ($P = 0.104$, $r = 0.22$) to assess the association between bacterial community structure and soil environmental factors by canonical correspondence analysis (CCA) using the vegan package in R.

Differentially abundant features were determined by Linear discriminant analysis Effect Size (LEfSe) in biobakery (*Segata et al., 2011*). Additionally, one-way analysis of variance (ANOVA) was applied to evaluate significant differences based on Least Significant Difference (LSD) method in species composition and soil chemistry among samples, with a least significant difference ($P < 0.05$) using SPSS 19.0 (IBM Corporation, Armonk, NY, USA).

# RESULTS

## Soil physicochemical characteristics

Soil physicochemical characteristics potentially affected by *Eucalyptus* plantations are shown in Table 1. The soil pH and SOC in the secondary forest and 5YR *Eucalyptus* soil were higher ($p < 0.01$) than 2YR and 10YR *Eucalyptus* soils. By contrast, available phosphorus (AP) of 2YR, 10YR *Eucalyptus* soil were higher than 5YR and secondary-forest samples ($p < 0.05$), and TP increased significantly ($p < 0.01$) with the time of *Eucalyptus* planting, while reached a stable value between 5YR and 10YR soils. The content of TK decreased with plantation age of *Eucalyptus*, but recovered higher than origin in the later plantation stage. The $NH_4^+$-N content was significantly different ($p < 0.01$) comparing the different age of *Eucalyptus* soil and secondary forest; the lowest value was in the 10YR

Li et al. (2018), *PeerJ*, DOI 10.7717/peerj.5648

**Table 1  Physicochemical properties in *Eucalyptus* plantations of different ages and secondary forest.**

| Sample | pH | SOC (g/kg) | TN (g/kg) | TP (g/kg) | TK (g/kg) | AN (mg/kg) | NH4 (mg/kg) | NO3 (mg/kg) | AP (mg/kg) | AK (mg/kg) |
|---|---|---|---|---|---|---|---|---|---|---|
| SF | 4.34 ± 0.09a | 60.32 ± 12.05a | 2.14 ± 0.62a | 0.22 ± 0.03c | 13.8 ± 1.77ab | 150.28 ± 43a | 21.82 ± 2.64b | 7.32 ± 1.59b | 2.98 ± 1.78b | 100.32 ± 57.11a |
| 2YR | 4.01 ± 0.06b | 50.57 ± 6.4b | 1.74 ± 0.04a | 0.27 ± 0.01b | 11.67 ± 2.04b | 143.22 ± 2.94a | 15.87 ± 3.81c | 22.73 ± 9.45a | 7.17 ± 0.91a | 63.83 ± 7.14a |
| 5YR | 4.29 ± 0.12a | 61.92 ± 2.47a | 2.15 ± 0.13a | 0.32 ± 0.01a | 8.2 ± 0.42c | 164.58 ± 25.23a | 32.8 ± 2.98a | 10.37 ± 4.68ab | 3.15 ± 0.59b | 55.51 ± 11.63a |
| 10YR | 4.06 ± 0.07b | 42.41 ± 6.16b | 1.58 ± 0.16a | 0.32 ± 0a | 15.08 ± 0.52a | 133.12 ± 19.52a | 8.85 ± 2.35d | 21.21 ± 8.51a | 5.62 ± 0.54a | 91.63 ± 6.39a |

**Notes.**

Different letters in rows indicate significant difference between the samples at $P < 0.05$, $n = 3$.

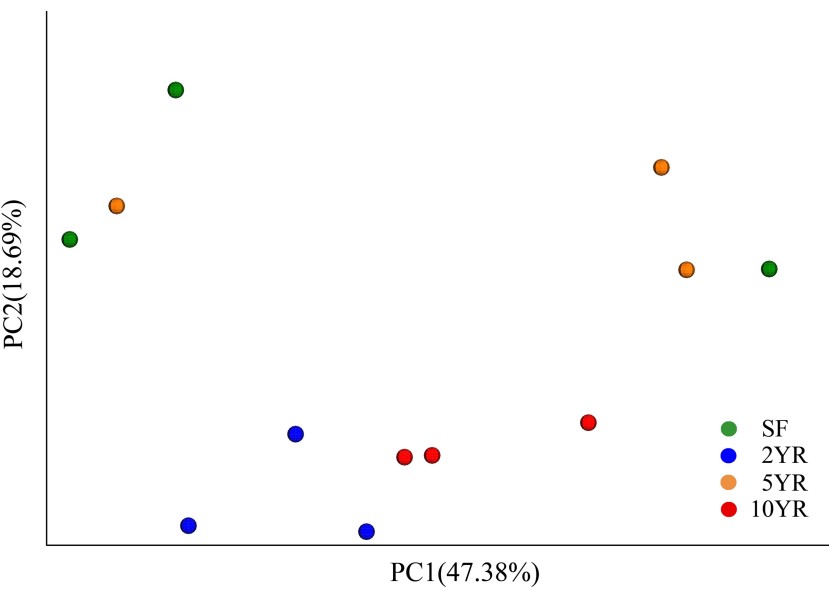

**Figure 2** Principal coordinates analysis of bacterial community composition (weighted UniFrac dissimilarity) in *Eucalyptus* plantations and secondary forest soil.

soils. But there were no significant changes in soil TN, AN and AK contents among the secondary forest and *Eucalyptus* plantations.

## Diversity of soil bacterial community

In total, 427,687 high-quality sequences were obtained from 12 soil samples in three different age stages of *Eucalyptus* plantations and secondary-forest with reads quality trimmed to 270 bp. subsequent reads were normalized to 27838 sequences with rarefaction curves (Fig. S1) determining that this depth was sufficient to describe the sample diversity.

The numbers of observed sOTUs were 648,642,684,643 in 2YR, 5YR, 10YR *Eucalyptus* plantation and secondary-forest respectively (Fig. S1). The estimated Shannon indexes did not significantly differ among different ages of *Eucalyptus* plantation and secondary forest (Fig. S2). The Chao1 index increased along the *Eucalyptus* plantations however this trend was not statistically significant (Fig. S2).

## Structure and composition of bacterial communities

When using weighted Unifrac Samples formed distinct clusters in the PCoA plot (Fig. 1) based on planting year, one cluster consisting of samples from 2 year and 10 year plantations and one with samples from 5 year and SF plantations (Fig. 2). This however was primarily explained by axis 2 (18.69%) with grouping being less clear along the primary axis, which explained 47.38% of variation. Interestingly, when we used unweighted Unifrac samples grouped more clearly by plantation year indicating that diversity is to some degree partitioned by age, however this is primarily being realized in less abundant taxa (Fig. S3). Further statistical analysis revealed that overall, plantation year did not significantly explain beta-diversity patterns (anosim result, $P = 0.073$), however the difference between

**Table 2  Results of mantel test between different soil properties and community composition.**

| Soil properties | r | P-value |
|---|---|---|
| pH | 0.328 | 0.025 |
| TP | 0.303 | 0.041 |
| TN | 0.297 | 0.048 |
| AP | 0.22 | 0.104 |
| NH4 | 0.119 | 0.387 |
| NO3 | 0.114 | 0.406 |
| AN | 0.094 | 0.541 |
| SOC | 0.085 | 0.556 |
| AK | 0.032 | 0.863 |
| TK | −0.012 | 0.913 |

**Notes.**
"*r*" is symbol for Spearman rho.

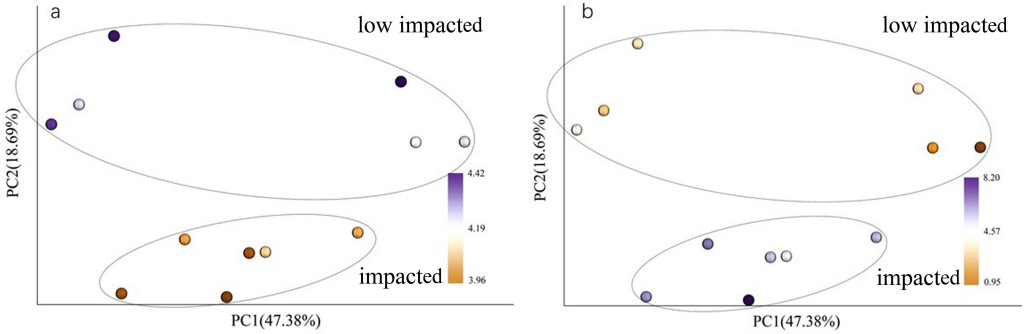

**Figure 3  PCoA with color gradients mapped to (A) pH and (B) AP.** Circles denote clusters defined as impacted and low impacted.

the 2YR/10YR cluster and the 5YR/SF cluster was significant ($P = 0.014$). As the 2YR and 10YR samples strongly differ from the control non-plantation SF samples, we have termed this cluster "impacted". However, as the community in the 5YR samples groups with the control SF plantation we have termed this group "low impacted" (refer to Discussion). Mantel tests showed that pH and AP were primary environmental drivers of beta-diversity patterns (Table 2) and clearly differ between the 5YR/SF low impacted cluster and 2YR/10YR impacted cluster (Fig. 3).

Sequences variants were classified into 29 phyla, 85 classes, 150 orders, 236 families and 344 genera, including some unclassified or no rank species. The dominant phyla included *Proteobacteria*, *Actinobacteria*, *Acidobacteria*, *Chloroflexi*, *Planctomycetes*, *WPS-2* and *Verrucomicrobia* (Fig. 4), whose relative abundance were above 2%. The *Proteobacteria* had the highest abundance, especially in secondary forest, but decreased with the increased years of *Eucalyptus* plantation, accounting for 44.22%, 41.3%, 39.75% and 37.34%, respectively (Fig. 4), however this difference was not statistically significant (ANOVA $P > 0.05$, Table 3). The relative abundance of *Actinobacteria* increased significantly (ANOVA $P < 0.05$) in 2YR *Eucalyptus* plantations, but recovered to the SF abundance

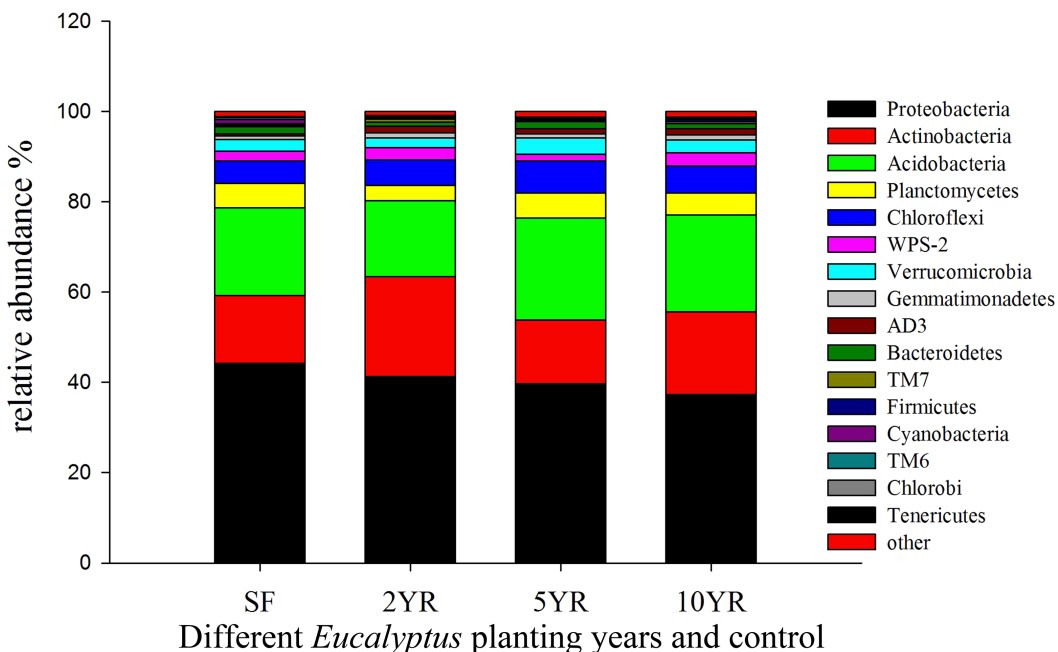

**Figure 4** Relative abundance of bacterial phyla (three samples including in each treatments).

in later plantation years (Fig. 4, Table 3). *WPS-2* climbed significantly in 2YR and 10YR but was detected no significant variation in 5YR and secondary-forest (Fig. 4, Table 3). In comparison with the secondary-forest soil, the relative abundance of *Cyanobacteria*, *TM6* and *Tenericutes* in *Eucalyptus* forest showed a significant decrease (ANOVA $P < 0.05$), but not with the increasing year of *Eucalyptus* plantation relative to SF. However, the *Gemmatimonadetes* and *AD3* were significantly higher in *Eucalyptus* plantation than that in secondary forest (ANOVA $P < 0.05$, Fig. 4, Table 3).

At the genus level (Table S1, approximately 50% were listed), the dominant genera in both *Eucalyptus* plantation soil and secondary forest were *Rhodoplanes*, and lineages within the order *Actinomycetales* and family *Rhodospirillaceae*, however these showed no significant variation between samples. Abundant genera (>2%) displaying significant differences in abundance between samples based on one-way ANOVA ($P < 0.05$) were listed in Table S1. *Candidatus Koribacter* increased in abundance in the 10YR (1.94%) sample relative to SF, however did not significantly change in the three treatment groups. The relative abundance of *Bradyrhizobium* showed a significant drop in the 10YR (1.86%) sample, but was consistent across early stages of *Eucalyptus* plantation. Additionally, the *Conexibacter* in 5YR (0.14%) decreased significantly compared with the 2YR (20.48%) sample, although was not significantly different between *Eucalyptus* samples and SF. By contrast, *Skermanella* in the 5YR (0.13%) soil decreased relative to SF (1.21%), but recovered in 10YR (0.53%) samples. *Azospirillum* increased significantly in 2YR *Eucalyptus* samples (0.98%), while relative abundance decreased back to the original level in 5 and 10 year plantations.

**Table 3** Differences in relative abundance of bacterial phyla in different stages of *Eucalyptus* plantation and secondary forest.

| Phylum/Treatment | SF | 2YR | 5YR | 10YR |
|---|---|---|---|---|
| *Proteobacteria* | 44.22 ± 10.95a | 41.3 ± 2.21a | 39.75 ± 8.33a | 37.34 ± 1.48a |
| *Actinobacteria* | 15.04 ± 3.63b | 22.18 ± 2.38a | 14.09 ± 3.49b | 18.37 ± 4.15ab |
| *Acidobacteria* | 19.47 ± 6.67a | 16.81 ± 2.07a | 22.56 ± 4.54a | 21.4 ± 1.76a |
| *Planctomycetes* | 5.38 ± 2.7a | 3.33 ± 1.13a | 5.51 ± 2.16a | 4.79 ± 1.09a |
| *Chloroflexi* | 5.03 ± 2.47a | 5.72 ± 0.31a | 7.22 ± 2.78a | 6.07 ± 1.84a |
| *WPS-2* | 2.1 ± 0.09b | 2.75 ± 0.17a | 1.41 ± 0.56b | 2.9 ± 0.28a |
| *Verrucomicrobia* | 2.64 ± 1.69a | 2.09 ± 0.96a | 3.6 ± 1.86a | 2.87 ± 0.16a |
| *Gemmatimonadetes* | 0.75 ± 0.21b | 1.11 ± 0.04a | 0.92 ± 0.01a | 1.13 ± 0.18a |
| *AD3* | 0.40 ± 0.25b | 1.49 ± 0.46a | 1.13 ± 0.20a | 1.38 ± 0.68a |
| *Bacteroidetes* | 1.57 ± 0.85a | 0.9 ± 0.07a | 1.58 ± 0.75a | 1.05 ± 0.15a |
| *TM7* | 0.39 ± 0.1b | 0.71 ± 0.12a | 0.31 ± 0.08b | 0.52 ± 0.19ab |
| *Firmicutes* | 0.33 ± 0.17a | 0.2 ± 0.08a | 0.21 ± 0.1a | 0.44 ± 0.23a |
| *Cyanobacteria* | 0.96 ± 0.63a | 0.29 ± 0.10b | 0.23 ± 0.12b | 0.23 ± 0.04b |
| *TM6* | 0.36 ± 0.13a | 0.16 ± 0.04b | 0.24 ± 0.09b | 0.16 ± 0.05b |
| *Chlorobi* | 0.01 ± 0.02b | 0.05 ± 0.04ab | 0.03 ± 0.02ab | 0.07 ± 0.03a |
| *Tenericutes* | 0.27 ± 0.02a | 0.00 ± 0.01c | 0.07 ± 0.05b | 0.02 ± 0.02bc |

**Notes.**
Different letters in rows indicate significant difference between the samples at $P < 0.05$, $n = 3$.

Overall 16 taxonomic groups were overrepresented in samples forming the impacted cluster and 19 taxonomic groups were overrepresented in the low impacted cluster samples (Fig. 5) as determined by LEfSe LDA scores (Fig. S4). In particular, taxa at different phylogenetic levels within the *Actinobacteria* phylum were significantly different in abundance between the impacted and low impacted clusters (Fig. 5). These included the *Thermoleophilia* order, *Acidimicrobiia* order, *Solirubrobacterales* class, *Acidimicrobiales* class, *Pseudonocardiaceae* family and *Nocardiaceae* family. Moreover, At the genus level, *Bradyrhizobium* were found to predominate in soils of low impacted cluster, as well as *Myxococcales* at the order level. The plots of features above with statistically significant differences between clusters were put into supplementary materials (Fig. S5; Fig. S6; Fig. S7).

## Relationships between bacterial community and soil characteristics

CCA (Canonical Correlation Analysis) was carried out to identify the main soil characteristics driving community patterns in *Eucalyptus* plantation and secondary forest soil samples. According to the Mantel test results (Table 2), four soil properties demonstrated a strong relationship to beta diversity patterns and are included as vectors in the CCA analysis (Fig. 6). The first two axes of the CCA explain 31.14% and 26.58% of the variability, respectively (Fig. 6). The result showed that pH was positively associated with 5YR and SF soil samples (the low impacted cluster), but negatively associated with 2YR and 10YR and explained 89.66% of CCA1. In addition, likewise on CCA1, TP and AP were positively correlated with the 2YR and 10YR soils (impacted cluster), contrary to the 5YR and SF samples.

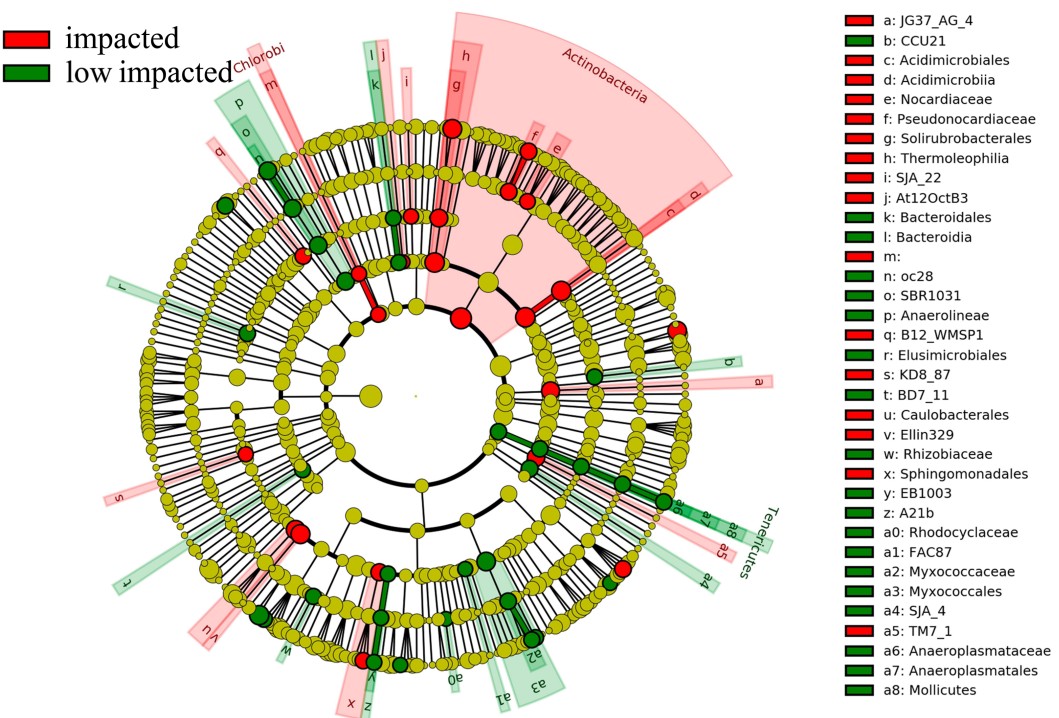

**Figure 5** Differently abundant taxa abundance shown within phylogenetic lineages (LEfSe analysis) between impacted (comprised by 2YR and 10YR) and low impacted (comprised by 5YR and SF) clusters.

## Function prediction of bacteria

The potential metabolic functions of soil taxa were predicted using PICRUSt, a software package that uses the genomic composition of organisms closely related to the 16S rDNA composition of the sampled community and the predicted metagenome correlated to an actual metagenome with a Spearman's R of 0.81 for soil microbial communities (*Langille et al., 2013*). Related genes collapsed into KEGG pathways at level 3 of the KEGG hierarchy with a mean NSTI score of 0.12 and varied significantly between clusters of *Eucalyptus* plantation with 20% of pathways having a corrected *P*-value <0.05 (Table S2).

In particular nitrogen metabolism, amino acid metabolism and energy metabolism, degradation in bisphenol, chlorocyclohexane and chlorobenzene, chloroalkane and chloroalkene decreased significantly in the impacted cluster (Fig. 7, Table S2). Additionally, glycoysis and TCA cycle in impacted increased significantly than that in low impacted samples.

## DISCUSSION

Overall what makes this study site, a model system for the influence of *Eucalyptus* plantation, interesting is that during a chronosequence of *Eucalyptus* plantation, composition and function of soil bacteria community were shifted in impacted years but lower impact was

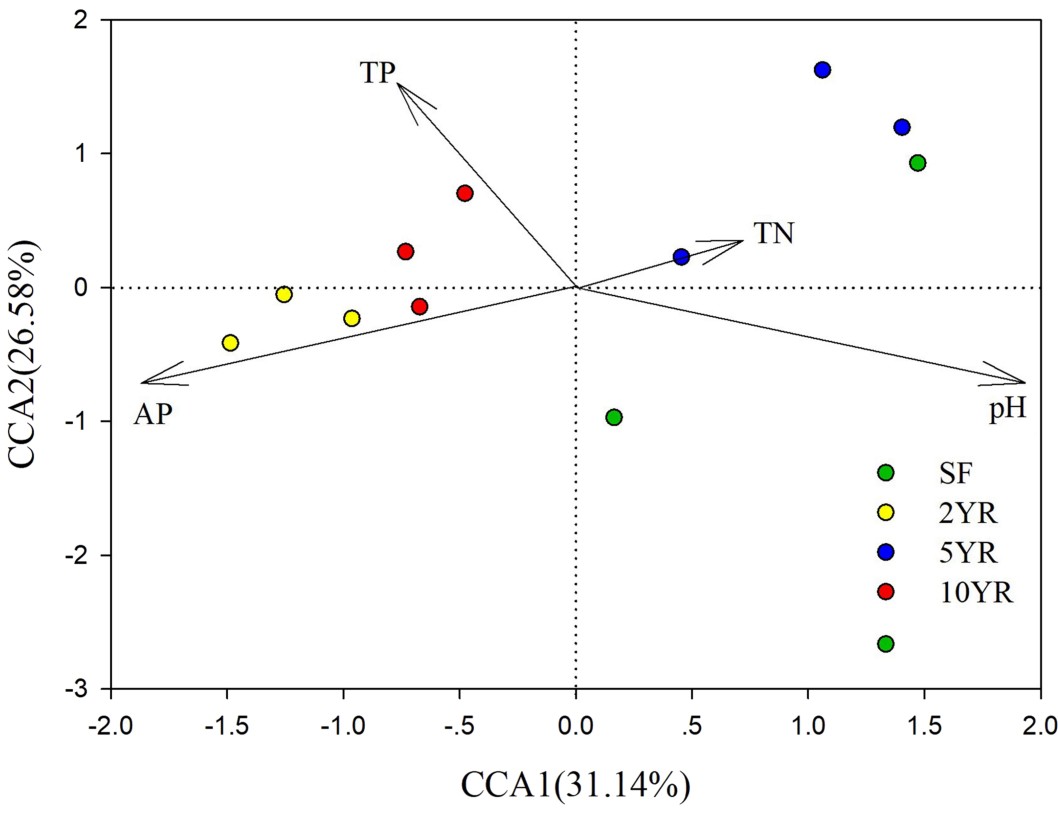

**Figure 6** Canonical Correlation Analysis (CCA) of the abundance of sOTUs in bacterial community and soil environmental variables of the *Eucalyptus* and secondary forest.

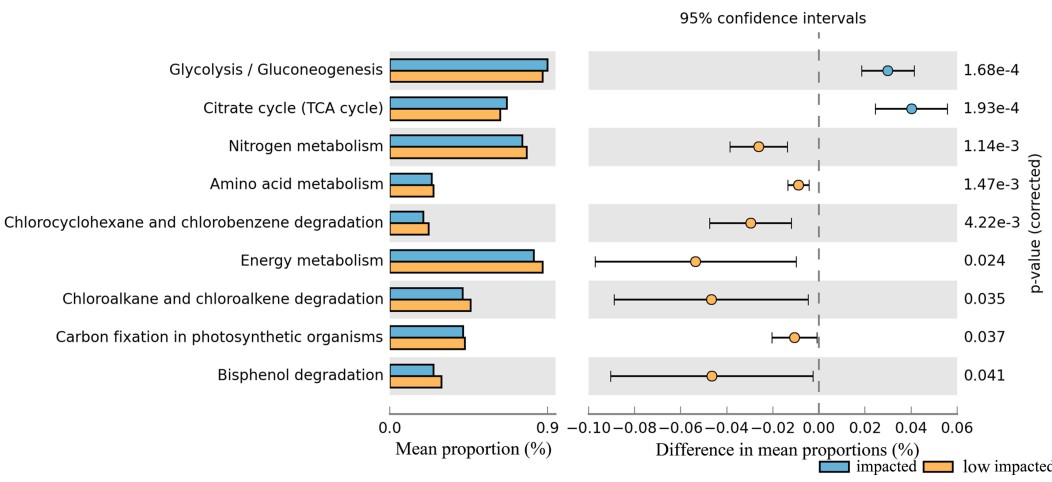

**Figure 7** Relative abundance of predicted soil bacterial functions in impacted and low impacted clusters predicted by PICRUSt using KEGG Orthologs. Pathways presented here are relevant to soil ecosystem function and compound degradation, full data supplied in Table.

observed in the 5-year plantation. Furthermore, soil properties were altered significantly, which were treated as a main factor driving community variation.

## Soil properties

Key soil properties such as pH, SOC and phosphorus concentration (TP and AP) were altered between secondary forest and *Eucalyptus* plantations or shifted with plantation time indicating the influence of *Eucalyptus* on resident soil microbial communities. Soil pH as an indicator of soil condition is affected easily by understory vegetation and plantation. *Eucalyptus* plantations generally may lead to soil acidification when converted from original vegetation, as observed after ten years (*Rhoades & Binkley, 1996*; *Sicardi, Garciia-Préchac & Frioni, 2004*). In our study, the lower pH values were shown in 2YR and 10YR *Eucalyptus* soil compared with secondary-forest, which may be a consequence of the accelerated extraction of cations and of compounds released from decaying leaf litter (*Soumare, Sall & Sanon, 2016*). Consistent with our results, the amount of soil SOC has been shown to decrease after planting *Eucalyptus* (*Behera & Sahani, 2003*; *Cook, Binkley & Stape, 2016*; *Zhang et al., 2015*) but also to increase in some conditions (*Zhang et al., 2012*). Infertile soil organic matter may decrease soil structural stability leading to erosion (*Behera & Sahani, 2003*). The investigation of soil properties indicated that soil SOC content in 2YR and 10YR was less than that in secondary-forest. Nevertheless, the highest soil SOC value in 5YR *Eucalyptus* plantation may be caused by low decomposition rates of microbial communities (*Zheng et al., 2017*). Soil SOC and other soil nutrients differed between *Eucalyptus* soil and original forest, such as TP and AP, or stabilized under certain conditions (TP), probably partly arising from processes such as litter fall decomposition, leaching and mineralization (*Falkiner & Smith, 1997*; *Jaiyeoba, 1998*; *Laclaua et al., 2003*). Phosphorus concentrations may exhibit an increase during the process of decomposition of leaf litter (*Ribeiro, Madeira & Araújo, 2002*; *Han et al., 2011*).

## Community diversity patterns

Several of the above variables were identified as key drivers in determining microbial community structure and underpinning the clustering of samples into groups determined by plantation age. pH has been identified as an important variable in determining microbial diversity patterns on a number of scales (*Fierer & Jackson, 2006*) and is a factor in determining plantation soil community composition (*Zhou et al., 2017a*; *Zhou et al., 2017b*). Nutrient concentration (P, N) is also generally a major driver of microbial diversity patterns (*Leff et al., 2015*) including in forest habitats (*Liu et al., 2012*) and *Eucalyptus* plantations (*Lan et al., 2017*) and our results are consistent with this.

The low explanatory power of CCA1 and CCA2 may suggest that some relative factors were not measured and that these also play a role in determining community diversity patterns.

Overall, our beta-diversity patterns showed a cycle of impact (2YR/10YR), which was defined as a strong dissimilarity to the native soil conditions in the control SF, and low impacted (5YR), where soil microbial diversity patterns resolved to a high-degree of similarity to control SF samples. This was not stable over time as samples were low impacted

after 5 years then displayed community profiles indicative of higher impact after 10 years. The lower impact of soil communities has been observed following *Eucalyptus* plantations recently (*Chen et al., 2013*) however short-term fluctuations in microbial diversity are not well described in plantation systems. The mechanism for this is potentially driven by patterns mediated by a lowering of soil pH and increases in nutrient concentration as a consequence of leaf litter and tree-root inputs, such as allelochemicals, as described above (*Khan, Khitran & Baloch, 1999*; *Mitchell et al., 2012*) in the impacted samples. Indeed, following experimental addition of *Eucalyptus* leaves to control SF soil nutrient concentrations increased however this experiment was only conducted on a short time-scale (<6 months, Table S3). The 5 year samples, being defined by a higher pH and lower nutrient concentrations, were potentially less influenced by inputs derived from plant material. Collectively this highlights the importance of managing leaf litter, tree harvesting and nutrient input in influencing soil chemical properties and microbial community composition and that microbial responses to plantation are not stable over time.

Alpha diversity of soil bacterial community in *Eucalyptus* plantations has been found to be higher than native forest environment (*Silveira et al., 2006*; *Lan et al., 2017*). However, we found no significant differences in bacterial alpha-diversity (species richness, diversity indices) between *Eucalyptus* and other forest sites even though there were alterations in the structure of soil bacterial community. Thus the above inputs drive shifts in relative abundance rather than presence absence of community members. This homogeneity of alpha-diversity has been observed in similar plantations (*Kerfahi et al., 2016*).

## Taxonomic composition shifts

The most abundant phyla, *Proteobacteria*, *Actinobacteria*, *Acidobacteria* and *Verrucomicrobia* are well-represented in most forest soils (*Ederson, Terence & Tiedje, 2009*; *Fierer, Bradford & Jackson, 2007*; *Lan et al., 2017*; *Spain, Krumholz & Elshahed, 2009*), and the distribution of dominant bacteria were stable after *Eucalyptus* plantation in our study. The dominance of *Proteobacteria* and *Acidobacteria* as generally observed in most of soils indicated low impact by land-use type (*Rampelotto et al., 2013*), which may explain the small degree of variation among treatments of both of these phyla. *Acidobacteria* have been reported in oligotrophic habitats with low nutrient concentrations, as well as a wide range of metabolic organic matter, low C mineralization rate and ability to tolerate fluctuations in adverse soil conditions (*Aislabie & Deslippe, 2013*; *Fierer, Bradford & Jackson, 2007*; *Rampelotto et al., 2013*). Since *Proteobacteria* and *Acidobabcteira* could be considered as indicators of soil trophic level organisms (*Smit et al., 2001*; *Zhang et al., 2017*), the change from secondary-forest to *Eucalyptus* or stand age of *Eucalyptus* generated little effect on declining soil nutrition and metabolizing organic resource within the 10-year *Eucalyptus* plantation. Bacteria belonging to the phylum *Actinobacteria* were more dominant in impacted soils. *Actinobacteria* have been widely reported as playing critical roles in exuding antibiotics and secondary metabolites (*Quirinoa, Pappasa & Tagliaferroa, 2009*; *Lauber et al., 2013*), and their increased abundance in impacted samples could have consequences on these processes. Degradation of SOC in soils could lead to the high level of *Actinobacteria* observed in impacted soils (2YR and 10YR), because of its higher ability to consume

organic carbon pools and its copiotrophs lifestyle (*Rampelotto et al., 2013*; *Zhang et al., 2017*; *He et al., 2012*). Significant variation in the genus *Bradyrhizobium* between impacted and low impacted clusters illustrates the low ability of nitrogen-fixing in the *Eucalyptus* plantation (*Silva et al., 2014*). Likewise, *Cyanobacteria* was beneficial to nitrogen fixation and stabilized structure of soil by potentially binding particles in the terrestrial ecosystems (*Eldridge & Leys, 2003*; *Bowker, Maestre & Escolar, 2010*). Hence, it was probable that soil in *Eucalyptus* plantations had less tolerance from wind and water erosion than in the secondary-forest. Accordingly, we suggest that land-use altered the soil structure and composition, contributing toward the productivity of soil. Combined, the shifts in abundance of taxonomic groups with functional significance highlight the impact of *Eucalyptus* plantation on local soils.

### Shifts in functional potential

Taxonomic shifts between samples were reflected in the shifts in metabolic gene potential between the impacted and low impacted group, the result of which suggested a decrease in overrepresented functions in 2YR and 10YR impacted soils compared to low impacted potentially leading to an accumulation of metabolic products and nutrients. According to the gene families identified by PICRUSt analysis, we hypothesize that a decline in content of soil organic matter caused by breaking up macro-aggregates (*Hoosbeek et al., 2006*) likely triggered the increase of soil bacterial capacity to fix carbon when artificial plantation was sustained for a few years, which may partially explain microbes impacted by *Eucalyptus* plantings (*Chen et al., 2013*; *Cortez et al., 2014*). Overall, shifts in microbial function were evident between clusters indicating that pH and nutrient shifts as a consequence of plantation will likely impact microbial function in these soils.

## CONCLUSIONS

Within a chronosequence of *Eucalyptus* plantation, soil microbial community structure shifted significantly in soils 2 and 10 years after plantation compared with secondary forest. Following 5 years of plantation, low impact was observed with the community showing a high-degree of similarity to control soils. The main factors which drove this partitioning were variation in pH and nutrient concentrations, such as phosphorus potentially resultant from leaf litter and plant inputs. Microbial communities were not stable over the time-scale measured, highlighting the need to understand microbial responses to plantation over varying time-scales to manage ecological outcomes for plantations, such as stable, healthy soil microbial communities.

## ACKNOWLEDGEMENTS

We would like to thank Lin Huang and Hui Wang for their generous help in soil sampling.

### Funding

This work was financially supported by the Project of the Department of Forest of Guangdong Province (Grant No. YUE CAI NONG [2017] No. 83) and the Project of State Forestry and Grassland Administration. The funders had no role in study design, data collection and analysis, decision to publish, or preparation of the manuscript.

### Grant Disclosures

The following grant information was disclosed by the authors:
Project of the Department of Forest of Guangdong Province: YUE CAI NONG [2017] No. 83.
Project of State Forestry and Grassland Administration.

### Competing Interests

The authors declare there are no competing interests.

### Author Contributions

- Jiayu Li conceived and designed the experiments, performed the experiments, analyzed the data, prepared figures and/or tables, authored or reviewed drafts of the paper, approved the final draft.
- Jiayi Lin contributed reagents/materials/analysis tools.
- Chenyu Pei performed the experiments.
- Kaitao Lai analyzed the data, contributed reagents/materials/analysis tools.
- Thomas C. Jeffries analyzed the data, prepared figures and/or tables, authored or reviewed drafts of the paper, approved the final draft.
- Guangda Tang conceived and designed the experiments, contributed reagents/materials/analysis tools, authored or reviewed drafts of the paper, approved the final draft.

### DNA Deposition

The following information was supplied regarding the deposition of DNA sequences:
The raw data is available at the European Nucleotide Archive database under accession code ERP109013.

### Data Availability

Li, Jiayu (2018): rawdata.zip. figshare. Dataset. https://doi.org/10.6084/m9.figshare.6275183.v1.

### Supplemental Information

Supplemental information for this article can be found online at http://dx.doi.org/10.7717/peerj.5648#supplemental-information.

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
