# Peer review of "Variation of soil bacterial communities along a chronosequence of Eucalyptus plantation"

_PeerJ, doi:10.7717/peerj.5648_

## Round 0.1 · original submission · Major Revisions

Please pay close attention to the comments about fully explaining the experimental design. Also, please clearly address the comments about the validity of the design for testing your question brought up by the second reviewer; you may want to consider re-framing your work so that the research question addressed by the paper better matches the study design. Your resubmission will likely be sent for re-review.

Reviewer 1 ·

Basic reporting

Please see general comments

Experimental design

Please see general comments

Validity of the findings

Please see general comments

Additional comments

Li and colleagues have analysed soil samples from a chronosequence of Eucalyptus forests, as well as a secondary forest, to determine the changes that occur in bacterial community structure and inferred function. The authors found that the soil communities under two and 10 year old eucalyptus plantations were significantly different to those found under a five year old eucalyptus plantation, and a secondary forest. This difference could be linked to differences in soil chemistry, and was also reflected in differences in the inferred function of microbial communities.

This paper has the potential to provide useful insights into the microbial communities residing in Eucalyptus forest soils. However, the description of the sampling design (lines 84-92) is very confusing, and does not clearly indicate how many sites were sampled, or the spatial scale covered. While 12 samples in total is a small dataset regardless, if these 12 samples are from separate sites (meaning n=3 replicates sites per forest type), the data could be meaningfully interpreted. However, if only four sites were sampled in total (meaning n=1 per forest type), this would be inadequate to draw the conclusions presented in this manuscript. The authors must clarify this important point.

Additionally, classifying the 2YR and 10YR plantation forests as ‘impacted’ and the 5YR and SF as “recovered” does not make sense (lines 177-185; 225-227). How can five year old Eucalyptus plantations be considered recovered if they are then considered “impacted” again five years later? The terminology used for these two groups should be changed, and accordingly the word “recovery” should be removed from the title of the manuscript. Additionally, since it is unclear if each sample within a forest group is from the same site, or from different sites, it is difficult to judge whether adequate sampling has been done to determine when a Eucalyptus plantation is different to a secondary forest.

The manuscript can be further improved in these areas:
Lines 58-65: There are examples of other relevant studies that do use NGS techniques and these should be mentioned here. For example Lan, G. et al., (2017) Soil Bacterial Diversity Impacted by Conversion of Secondary Forest to Rubber or Eucalyptus Plantations: A Case Study of Hainan Island, South China. For. Sci. 63: 87–93.
Line 73: “nutrient cycling and metabolic etc.” Should this say metabolism? The use of etc. also makes it a vague statement and should be better defined as it is a main research question.
Lines 84-85: How far apart was each replicate plot and what is meant by “S” shape? 20 x 20 cm is a very small “plot” size, how many samples were collected within each of these plots? Perhaps a figure showing the sampling design could be useful.
Line 87-88: How have six replicate plots, in four forest types resulted in 12 composite samples? How much distance separated each forest site? How many replicates were sampled for each plantation stage/secondary forest?
Line 106-108: These primers amplify the V3-V4 region, not just V4. Please correct this.
Line 110-113: Please indicate the sequencing length, and number of sequencing cycles that were used.
Line 119: Please explain what is meant by sOTU, as it is not commonly used. My understanding is that the output from DADA2 is a table of ASV (amplicon sequence variants) so why has this terminology not been used instead?
Lines 131-132: Please clarify if a rarefied OTU/ASV table has been used only for alpha-diversity analyses, or for all analyses.
Lines 140-142: The justification of the variables included for this analysis does not make sense. Both AP and AK were not significant according to the Mantel test, so why are they included? Furthermore, Mantel R values should be presented along with the p-values, or direct the reader to Table 2.
Lines 150-159 and lines 252-271: The detailed descriptions of differences in soil chemistry, without relating these directly to bacterial data, seems unnecessary as it was not a main research goal. I would remove these two passages.
Line 165: The number of OTUs per sample seem very low compared to what is normally reported for soil bacterial communities. Were low abundant OTUs/ASVs removed, or any other filtering steps included? This should be noted in the methods.
Lines 177-185; 225-227: As discussed above, the use of “recovered” and “impacted” does not make sense for the data presented, please use alternative terminology.
Line 205: The table that is referred to here (Table S1) was not included in the review material.
Lines 215-222: For the specific taxa that are discussed, please include the histogram of relative abundances that is part of the LEfSE output (i.e. the plot in Figure 1c in the LEfSe paper: http://doi.org/10.1186/gb-2011-12-6-r60.) This should be in the supplementary information, but is important to determine the “biological relevance” of these results, and ensure they are not due to outliers.
Figures 1-2: The manuscript has a lot of figures, some of which are barely referred to. I recommend moving figures 1 and 2 into supplementary. ‘(P<0.05)’ should also be removed from Figure 2’s legend or put into context.
Figure 5: It would be helpful if the legend mentions that it is showing the abundance of different phyla for different forest ages/secondary forest, and how many samples were combined for each ‘bar’. There should also be an X-axis title.
Table 1: Please indicate what statistical test was used to obtain p-values, either in the table heading or in the methods. Please include standard error values to show variability among samples from the same forest type.
Table 3: Please indicate what statistical test was used to obtain p-values, either in the table heading or in the methods.
Figure S2: Some of the bars have no taxon names showing.
There is no data deposition statement in the manuscript, please provide this. The EBI accession number provided with the manuscript submission details is not yet accessible, but raw data were available for review through figshare.

There are several grammatical and spelling errors throughout the manuscript, along with referencing issues; some of these are listed below, but this is not an exhaustive list and the manuscript could benefit from further proof reading:
Line 42: The reference includes the lead author’s first and last name, and this is also incorrect in the reference list
Line 51: remove the word “which”
Lines 52-55: The two statements in this sentence are somewhat contradictory. Please explain what is meant by not “negatively affected” if there is evidence of differences in community structure
Line 78: Should read “on June 8th” not “in June 8th”
Line 133: Incorrect spelling of Kruskal-Wallis
Line 139: Reference is in all capitals.
Line 198: remove “relative to SF” at end of sentence
Line 216: overrepresented, not “overpresent”
Line 304: ‘of evenness’ should be deleted
Line 316: Remove “of” after “Since”
Line 323-325: This doesn’t make sense, please clarify
Line 327: The reference includes the lead author’s first and last name, please correct.

Reviewer 2 ·

Basic reporting

This manuscript by Li et al. entitled "Recovery and variation of soil bacterial communities
along with a chronosequence of Eucalyptus plantation" describes the effects of Eucalyptus plantation on soil and in bacterial soil inhabitants. In short, the authors study how the community would change and recover in the chronosequence of Eucalyptus plantation. In conclusion, the authors show that soil bacterial community change after 2 years of Eucalyptus plantation, recover after 5 years and change again after 10 years. Overall, the manuscript is average written, descriptive, and although contains results of interest, the conclusions drawn are not supported by the results.

Experimental design

While reading and evaluating this manuscript, a variety of questions and remarks came regarding the experiment. More information is needed related to sampling site and just a few replicates are available to draw strong conclusions related to the recovery and variation of the bacterial community. However, it is an original research and the molecular, bioinformatic and statistical analysis was described with sufficient details.

Validity of the findings

The results are well described but I am not convinced about the recovery and change of the bacterial community in the consequence. To have this conclusion, soil samples should be sampled from the same area and needed to be monitored over time. There is a novelty in the manuscript and data are robust (although from only 12 samples). However, the conclusions from the results are not supported.

Additional comments

the subject is relevant, however, the authors did not succeed to answer the research questions with the experiment. To track recovery of bacterial community, the community should have been assessed by comparing the soil samples before and after Eucalyptus plantation in the same plot. Their conclusions are not supported by the experiment.

---

## Round 0.2 · Minor Revisions

Please address all the minor comments raised by the reviewer.

Reviewer 1 ·

Basic reporting

See general comments

Experimental design

See general comments

Validity of the findings

See general comments

Additional comments

The revised manuscript submitted by Li and colleagues is a significant improvement on the original draft. I thank the authors for their careful consideration of my previous comments, and their responses.

The revised description of the sampling design highlights that this study was conducted by sampling only one plantation per group (or age). Therefore, the results from this study cannot be extrapolated beyond what is happening in these four sites, and it cannot be determined if the results hold true for other Eucalyptus plantations. While for the most part the authors do not over-state their results, I feel that there is not enough evidence in this paper alone for the recommendation in lines 359-362 and would recommend removing this to not “over sell” the results presented in the manuscript.

In my opinion, a few minor changes are required before the article is ready for publication:
Please specify the number of soil cores taken per plot. I.e. how many soil cores went into each composite sample (lines 87-89).
The authors have misunderstood my original comment asking them to include the sequencing length/number of sequencing cycles used. Illumina MiSeq chemistry can either sequence 2 x 250 bp or 2 x 300 bp fragments, depending on whether there are 500 or 600 cycles. Please remove “The number of sequencing cycles was 28” in lines 111-112, as this is the number of PCR cycles, and add the correct information to lines 112-114 (again, 468 bp is the average length of your PCR product, not the sequencing length).
Lines 129 – 131 claim that the results from the QIIME 1 pipeline are similar to those from the QIIME 2 pipeline, however the location of the samples on the PCoA plot are quite different. Either remove this statement or provide statistical evidence for the similarity between the two beta-diversity outputs.
Line 135 should say “number of sOTUs”
Line 244 states that PICRUSt has a “predicted minimum accuracy of 85-90% for soil”. Please explain where these values came from. The Langille et al 2013 paper does not make this claim; the closest statement to this within the paper is that the predicted metagenome correlated to an actual metagenome with a Spearman’s’ R of 0.81.

The manuscript still contains grammatical errors which at times make it difficult to read. Further proof reading would greatly increase the readability. Several examples, as well as a few typos, are highlighted below.
Line 17: “comparing with” should say “compared to”
Line 56: “Other relevant study that…” should say “Another relevant study of” also tropical is misspelled.
Line 59: What is meant by “artificial forests”
The research questions in lines 68 and 70 should either have the question marks at the end removed, or start with “How do”
Line 79-80 is hard to understand
Line 96: K2Cr2O7 should have a capital K
Line 123: the “and” should be removed
Line 137: Replace “Equally, according to the” with “Using”
Line 187: Use “Impacted” instead of “impact”
Line 221: Remove period after Overall
Lines 248-251: There are multiple unnecessary capitals in this paragraph
Line 292-293: Again, this sentence is hard to understand, please clarify
Line 309: Replace “even though alterations” with “even though there were alterations” and remove occurring
Line 315: Remove “of”
Line 321: change “tolerance” to “tolerate”
Line 323: “an indicators of” should either be “indicators of” or “an indicator”
Line 334: Change “illustrates that low ability” to “illustrates the low ability”
Line 344: “suggested a decrease in overrepresented” overrepresented compared to what?

---

## Round 0.3 · accepted · Accept

Thank you for your full responses to reviewer comments. Please make one change before your final submission: combine the last sentence of the conclusion to the final paragraph.

#